# The Effect of the Chorion on Size-Dependent Acute Toxicity and Underlying Mechanisms of Amine-Modified Silver Nanoparticles in Zebrafish Embryos

**DOI:** 10.3390/ijms21082864

**Published:** 2020-04-20

**Authors:** Zi-Yu Chen, Nian-Jhen Li, Fong-Yu Cheng, Jian-Feng Hsueh, Chiao-Ching Huang, Fu-I Lu, Tzu-Fun Fu, Shian-Jang Yan, Yu-Hsuan Lee, Ying-Jan Wang

**Affiliations:** 1Department of Environmental and Occupational Health, College of Medicine, National Cheng Kung University, Tainan 70101, Taiwan; q781001@gmail.com (Z.-Y.C.); lavender6328@hotmail.com (N.-J.L.); seed123465@gmail.com (J.-F.H.); 2Department of Physiology, College of Medicine, National Cheng Kung University, Tainan 70101, Taiwan; drosouv@gmail.com; 3Department of Chemistry, Chinese Culture University, Taipei 11114, Taiwan; zfy3@ulive.pccu.edu.tw; 4Department of Food Safety/Hygiene and Risk Management, College of Medicine, National Cheng Kung University, Tainan 70101, Taiwan; winds031215@gmail.com; 5Department of Biotechnology and Bioindustry Sciences, College of Bioscience and Biotechnology, National Cheng Kung University, Tainan 70101, Taiwan; fuilu@mail.ncku.edu.tw; 6The Integrative Evolutionary Galliforms Genomics Research (iEGG) and Animal Biotechnology Center, National Chung Hsing University, Taichung 40227, Taiwan; 7Department of Medical Laboratory Science and Biotechnology, Medical College, National Cheng Kung University, Tainan 70101, Taiwan; tffu@mail.ncku.edu.tw; 8Department of Cosmeceutics, China Medical University, Taichung 40402, Taiwan; 9Department of Medical Research, China Medical University Hospital, China Medical University, Taichung 40402, Taiwan

**Keywords:** zebrafish embryos, acute toxicity, silver nanoparticles, chorion pore size, lysosomal activity, apoptosis

## Abstract

As the worldwide application of nanomaterials in commercial products increases every year, various nanoparticles from industry might present possible risks to aquatic systems and human health. Presently, there are many unknowns about the toxic effects of nanomaterials, especially because the unique physicochemical properties of nanomaterials affect functional and toxic reactions. In our research, we sought to identify the targets and mechanisms for the deleterious effects of two different sizes (~10 and ~50 nm) of amine-modified silver nanoparticles (AgNPs) in a zebrafish embryo model. Fluorescently labeled AgNPs were taken up into embryos via the chorion. The larger-sized AgNPs (LAS) were distributed throughout developing zebrafish tissues to a greater extent than small-sized AgNPs (SAS), which led to an enlarged chorion pore size. Time-course survivorship revealed dose- and particle size-responsive effects, and consequently triggered abnormal phenotypes. LAS exposure led to lysosomal activity changes and higher number of apoptotic cells distributed among the developmental organs of the zebrafish embryo. Overall, AgNPs of ~50 nm in diameter exhibited different behavior from the ~10-nm-diameter AgNPs. The specific toxic effects caused by these differences in nanoscale particle size may result from the different mechanisms, which remain to be further investigated in a follow-up study.

## 1. Introduction

The rapid development of nanotechnology has stimulated the use of nanomaterials in various fields, including medical imaging, new drug delivery technologies and a variety of industrial products [1]. According to the survey of “The Community Research and Development Information Service, CORDIS” [2], the global market for nanomaterials is estimated at approximately 11 million tons/year with a market value of EUR 20 billion and growing. Among them, AgNPs are the most widely used nanoparticles (NPs) in consumer applications and account for approximately 24% of the commercial products [3]. Due to their superior antimicrobial activity, AgNPs are used in medical products such as catheters, implants and other materials to prevent infection [4,5]. In addition, AgNPs are frequently used in clothing, the food industry, paints, household products and other fields [6,7,8]. Although AgNPs are widely used, exposure to humans and their release into the environment and accumulation in aquatic ecosystems has become a concern [9,10]. Nevertheless, the comprehensive biological and toxicological effects of AgNPs in humans and the environment has still not been studied in detail compared to their commercialization in various consumer and medical products [11]. Thus, it is important to analyze the effects of AgNPs on ecological systems as well as human health.

The toxicity of AgNPs has been investigated using a variety of model systems from mammalian cell lines to aquatic organisms and rodent species [12,13,14]. Despite the enormous amount of published research, the understanding of the mechanism of action is still hindered by the variability of the NP characteristics used in the different studies. These differences include the size, coating, surface interaction, contact time, bioaccumulation and transformation under realistic environmental conditions [15,16]. These factors may change the toxicity profile and the interpretation of the results [15]. Specifically, the toxicity of AgNPs in most living organisms is size- and coating-dependent. Particle size determines and influences the way that AgNPs enter cells and their toxic effects on organisms [17,18]. It has also been reported that AgNPs coated with sodium citrate were more toxic than those coated with polyvinylpyrrolidone [19,20]. Regarding the size-dependent toxicity, Gliga et al. reported that small AgNPs have faster rates of Ag dissolution and higher cytotoxic potential than large AgNPs in a human lung epithelial cell model [21]. Nonetheless, Kim et al. showed greater cytotoxicity of 100-nm-sized AgNPs compared to smaller-sized particles (10 and 50 nm) in MC3T3-E1 and PC12 cells [22]. Conflicting observations have also been reported by using zebrafish embryos as the testing model. Kim and Tanguay indicated that the smaller 20 nm AgNPs were more toxic than the larger 110 nm AgNPs, regardless of the presence of the embryo chorion and test media [23]. In contrast, when compared with smaller AgNPs (13.1 ± 2.5 nm), larger AgNPs (97 ± 13 nm) incite more striking size-, stage-, and dose-dependent toxic effects of AgNPs upon embryonic development [24,25,26]. Thus, a debate persists regarding the size-dependent toxic effects and underlying mechanisms triggered by AgNPs. 

The zebrafish (Danio rerio) embryo (ZFE) test is an appealing in vivo model to assess the hazards of both conventional chemicals and NPs in (eco)toxicology [27,28,29,30,31]. The ZFE, as a vertebrate model species, combines the advantages of rapid development and optical transparency, allowing for easy observations of phenotypic responses in the internal organs, including the brain, jaw, eye, heart, yolk sac, trunk, tail, and so on [23]. In addition, it has been considered an alternative testing model to animals, allowing for the observation of lethal, acute, and sublethal toxicological endpoints [32,33]. In addition, massive amounts of ZFEs can be generated rapidly at a very low cost, permitting them to serve as a high-throughput in vivo assay for the study of developmental processes upon exposure to nanomaterials [24,25]. To estimate the toxic potency of AgNPs, the uptake of metals by ZFEs and the underlying kinetic processes may play a pivotal role [24]. It is expected that the distribution of metal-based nanoparticles follows a temporal and spatial pattern [32,34], whereas little is known about the association of NPs and ZFEs with regard to the quantification of the internalized amounts and the visualization of particle distribution around the embryo. Basically, the chorion is considered to be a barrier to the entry of NPs into zebrafish embryos [35,36]. Although the pore size of chorion canals (approximately 0.6–0.7 μm) is larger than the size of the NPs, the effect of the chorion on NP transport and subsequent biological toxicity may become complicated when NPs agglomerate or interact with chorion surface proteins.

Single types of in vitro cell culture assays are widely used to study the cytotoxic effects of nanomaterials, which can overlook vital and specific cell−cell interactions [37]. We previously demonstrated that AgNPs can be taken into fibroblast cells through endocytosis. The internalized AgNPs eventually accumulated in lysosomes or autophagosomes. The mechanisms of AgNP-induced autophagy dysregulation could be mediated by activation of oxidative stress and endoplasmic reticulum (ER) stress signaling pathways [37,38,39]. When compared with cell culture models, the ZFE test can simultaneously study the effects of NPs on a wide variety of cells and detect all related pathways, including oxidative and ER stress, involvement of apoptotic pathways and disruption of autophagy regulatory signaling in developing embryos. Most impressively, toxicological outcomes obtained from zebrafish and/or their embryos may be extrapolated to human biology because of their 70% of DNA similarity to humans [40,41]. 

The objective of the present study was to explore the toxic effects and mechanisms of AgNPs at the organismal and cellular levels by using the ZFE model system. To perform exposure studies, we selected AgNPs of two different sizes (10 and 50 nm) with amine surface modifications prepared using the same synthetic pathway. The specific focuses of our study were designed to assess (1) the effect of the chorion on AgNP transportation, (2) size-dependent mortality and lethal dose, (3) morphological defects of the embryos through the early developing stage, and (4) oxidative stress, apoptosis and autophagy-related lysosomal activity through in vivo monitoring of specific biomarkers in embryonic tissues. Our results showed that larger-sized (50 nm) AgNPs can transport and distribute into embryos more efficiently than smaller (10 nm) AgNPs, leading to severe developmental toxicity by inducing ROS-mediated stress responses. These findings are essential for a better understanding of AgNPs in aquatic ecosystems and provide important underlying mechanisms for ecological risk assessments of AgNPs and other nanoparticles.

## 2. Results

### 2.1. AgNP Properties

Two different sizes of amine-modified AgNPs were used to investigate the toxic effects on zebrafish embryos. The physicochemical characteristics of the small and large AgNPs were fully characterized with respedt to their size, morphology, surface charge and composition, as summarized in Figure 1. Transmission electron microscopy (TEM) images indicated that the AgNPs appeared mostly spherical with a mean pristine particle diameter of 13.04 ± 1.3 nm (SAS) or 52.3 ± 6.3 nm (LAS) (Figure 1A,B). Dynamic light scattering (DLS) analysis of SAS (average 30.0 nm, Figure 1C) and LAS (average 55.6 nm, Figure 1D) showed narrow peaks, indicating that the SAS and LAS both possessed a homogeneous dispersion. The scanning TEM and electron dispersive X-ray (EDX) analyses revealed that except for TEM-coated grid materials (carbon and copper), the SAS and LAS were composed of silver, which indicates the high purity of the nanomaterials (Figure 1E,F). According to DLS, the hydrodynamic sizes of the SAS and LAS were larger than in the dry state, which might be attributable to the aggregation of some nanoparticles in solution, and the zeta potentials for both types of particles were positive in water. We also measured the dispersity value; in general, the smaller the value of dispersity, the more stable the nanosuspensions. In our results, the dispersities of SAS and LAS were 0.263 and 0.185, indicating that the particles were stable following dispersion. In addition, the synthesized SAS and LAS were analyzed using UV-vis absorbance spectrophotometry to confirm their properties. As shown in Figure 1G, the maximum absorbance of the SAS was lower (399 nm) compared to the LAS at 430 nm.

### 2.2. Distribution of the AgNPs in Developing Zebrafish Embryos 

To investigate the distribution of AgNPs in zebrafish embryos, we used rhodamine B isothiocyanate (R6G)-conjugated SAS and LAS to provide dynamic imaging of zebrafish embryos. Figure 2A indicates that R6G-SAS accumulated in the outer layer of the chorion of the zebrafish embryos at 24 hpf and 48 hpf. Conversely, the LAS-exposed group tended to penetrate into the zebrafish embryo at 48 hpf (Figure 2B). To demonstrate whether AgNPs can penetrate into the zebrafish embryo, zebrafish embryos were exposed to SAS and LAS at different concentrations at 24, 72, and 96 hpf. Atomic absorption spectroscopy (AAS) was employed to quantify SAS and LAS (Figure 2C–E) in the zebrafish. The embryos exposed to SAS and LAS significantly increased AgNPs accumulative levels. Interestingly, accumulation of LAS was significantly higher than SAS. At 72 hpf, the accumulative AgNPs dramatically dropped to 0.018 (SAS 1 μg/mL), 0.03 (SAS 1 μg/mL), 0.15 (SAS10 μg/mL) and 1.69 (LAS 10 μg/mL) due to dechorionation upon embryos hatching. The zebrafish embryos exposed to SAS and LAS showed a concentration-dependent increase, but there was more silver ion content in the zebrafish body after exposure to LAS. 

### 2.3. Evaluation of Zebrafish Chorion Pore Size after Exposure to Different Sizes of AgNPs

The chorion protects the embryo and is the first barrier to come into contact with the SAS and LAS upon exposure. Using scanning electron microscopy (SEM), we observed that the inner layer of the chorion appeared to have evenly sized pores and a smoother surface in the control group. Both the SAS and LAS groups showed agglomerates around the inner membrane. By amplifying the LAS group, we could more clearly observe the granular agglomerates around the inner membrane pores (Figure 3A). In addition, we used backscattered electrons (BSE) to analyze the differences in the sample surface composition and to observe the accumulation of LAS in the outer or inner layers of the zebrafish embryo. As shown in Figure 3B,C, there was significant LAS accumulation in the inner or outer layer of the chorion. Interestingly, only exposure of LAS revealed an enlarged pore size in the chorion inner layer (Figure 3D). In the results of the pore size measurement, SAS exposure did not increase the pore size of the chorion compared to the control group, and its pore size was 0.78 μm. Instead, LAS exposure enlarged the pore size of the chorion (0.94 μm). The pores of the chorion are necessary for oxygen and nutrient transportation from the outer aquatic environment to the embryo and for the elimination of waste. However, AgNPs may interfere with the material exchange between the inner and outer layers of the chorion, resulting in toxic effects to zebrafish embryos.

### 2.4. Identifying the Effects of Different Sizes of AgNPs on Embryo Mortality and Morphological Abnormalities 

According to previous studies, the particle size and surface area correlated with NP-induced toxic effects and interactions with living organisms. Therefore, we determined the effects of particle size on mortality and developmental toxicity. The mortality of the AgNP-treated zebrafish embryos is demonstrated in Figure 4A,B. The embryos were exposed to 0, 0.05, 0.1, 0.5, 0.75, 1, 10, and 100 μg/mL SAS or LAS and the cumulative mortality was recorded at 24, 48, 72, 96, and 120 hpf. SAS and LAS dose-dependently decreased the survival rate. The lethal concentration 50 (LC_50_) of the SAS and LAS showed similar results (Figure 4C). Compared with the SAS groups, the mortality of LAS was higher than that of SAS. To investigate developmental toxicity, we observed body length, head-trunk angle and malformation. The body length and head-trunk angle are important indices of developmental toxicity. We exposed the embryos with AgNPs both in deionized water (Figure 5A,B) and E3 medium (Figure 5C,D), a medium usually applied to developmental study, and the results demonstrated that LAS suspended in deionized water significantly decreased the body length (Figure 5A) and head-trunk angle (Figure 5B), whereas SAS and LAS suspended in E3 medium elicit mild or none toxic effects on body length and head trunk angle when compared with deionized water. Furthermore, SAS and LAS increased the phenomena of malformations, such as pericardial edema (PE), yolk sac edema (YSE), opaque yolk (OY), axial curvature (AC), and jaw (J) malformation (Figure 6A,B). The most commonly observed malformations include notochord malformation, yolk edema, axis malformation, and heart malformation. Quantification of the total specific malformation rate of LAS was higher than that of SAS (Figure 6B). Taken together, both SAS and LAS resulted in zebrafish embryo developmental toxicity, increased severe malformation phenotypes and stronger toxic effects were observed when embryos were exposed to concentrations of 10 and 100 μg/mL LAS and SAS. Moreover, LAS induced more severe malformations than SAS.

### 2.5. Quantification of ROS Expression and Damage to the Intestines 

Overproduction of ROS induces multiple adverse effects and activates various cellular toxic pathways, including lipid and protein peroxidation, DNA damage, activation of autophagy and cell death [42]. Evidence abounds that excessive ROS generation is an important mechanism induced by metallic NPs [3]. Metallic NP-induced ROS are correlated with physicochemical properties, such as particle size and shape [3]. However, the role of AgNP size on ROS production remains elusive. Therefore, we next determined whether SAS and LAS caused different levels of ROS in zebrafish embryos. As demonstrated in Figure 7A,B, the signals from ROS were mainly located in the intestines. In addition, both SAS- and LAS-treated Tg(IFABP:dsRed) fish embryos induced excessive ROS, and LAS induced higher ROS levels than SAS (Figure 7C,D). These results suggested that SAS and LAS increased ROS levels in the intestines and that LAS triggered greater ROS generation than SAS.

### 2.6. AgNP-Induced Zebrafish Embryo Lysosomal Activity and Apoptosis

The lysosome is a critical catabolic and anabolic mediator that can coordinate signals for the degradation of cellular components and cellular stressors [3,43]. Among the multiple functions of the lysosome, lysosomes are involved in the terminal degradation of organelles for autophagy. Therefore, lysosomal activity alterations are tightly related to normal autophagy functions. Recently, several studies have indicated that damaged lysosomes can be an emerging mechanism of nanotoxicity [44]. Therefore, we determined the effects of particle size on lysosomal activity and further conducted a TUNEL assay to detect apoptosis and the main AgNP-induced development defects. To investigate lysosomal activity, zebrafish embryos were treated with SAS and LAS until 48 hpf and were stained with Lysosensor. The results indicated that both SAS and LAS increased lysosomal activity, especially after the treatment with LAS at higher concentrations (Figure 8A,B). In addition, SAS and LAS increased TUNEL^+^ cells (Red) (Figure 8C,D). Most interestingly, the LAS caused a greater number of TUNEL^+^-stained cells than the SAS in the notochord of zebrafish. Our results suggest that SAS and LAS induced lysosomal activity as well as apoptosis in the stages of zebrafish development.

## 3. Discussion

Nanotoxicology is an interdisciplinary field that must bridge both the complex physical and chemical properties of nanoparticles and their interaction with biological systems. The extraordinary properties of nanomaterials are primarily attributed to their nanoscale structure, size, and shape [43]. However, these good benefits may become a serious issue in the application of nanomaterials and bring unknown safety considerations. In the present study, we explored and answered some questions regarding different sizes of nanoparticle-induced nanotoxicity. To evaluate the size-dependent toxic effects of AgNPs, many studies have recently observed the survival rate, malformation, hatching rate, and heart rate of zebrafish embryos after exposure to AgNPs [11,45]. First, we analyzed the survival rate of zebrafish embryos after exposure to SAS and LAS. The results of the LC_50_ calculated by the statistical software indicated that the LAS have the potential to cause higher toxicity (Figure 4C). Most studies have suggested that a smaller particle size creates a higher specific surface area, resulting in increased bioavailability or surface activity of the particles, which in turn leads to increased toxicity [46]. However, some studies have indicated that a larger particle size is highly toxic to zebrafish embryos. Tracking the distribution of AgNPs in zebrafish embryos, the larger-sized AgNPs (41.6 ± 9.1 nm) caused more chorion penetration through passive diffusion and Brownian motion than the smaller particle size (11.6 ± 3.5 nm) [25,46]. Like the human gastric/intestinal mucosal barrier, the chorion offers effective protection and acts as the first barrier against microorganisms, toxins, and other environmental stresses during zebrafish embryo development. Previous studies have shown that nanoparticles may adhere to mucins, causing an enlargement of the pore size with increased susceptibility to penetration from microorganisms [47,48]. However, smaller, charged nanoparticles may be repelled by the hydrophilic domains and will not be able to penetrate the mucosal layer [47,49,50]. Scanning electron microscopy revealed that both the SAS and LAS passed through the pores into the zebrafish embryo and adhered to the inner chorion (Figure 3A). Interestingly, the groups exposed to the LAS had more aggregation in the inner membrane. The results of the BSE experiment also clearly indicated LAS accumulation on the inner and outer layers of the chorion, and microorganisms were observed on both the inside and outside of the chorion (Figure 3B,C). It is possible that we observed more adverse effects from the LAS-exposed zebrafish embryos due to the enlarged pore size of the chorion, which resulted in the penetration of microorganisms and caused infection or toxicity to zebrafish embryos by interfering with the chorion protection. More interestingly, we found the increasing ROS level and lysosomal activity in zebrafish embryo exposed to AgNPs (Figure 7 and Figure 8A). Lysosome is the definitive antimicrobial organelle [51]. It stands in a crucial position in host–pathogen interactions, by being both targeted by pathogens and serving as a major mechanism for killing intracellular invaders [52]. Previous study indicated that the change of lysosomal enzyme activity may control microbial invaders in infected prawns [53]. The induction of ROS and lysosomal activity in zebrafish embryo could be attributed, on one hand, by the increasing pore size caused by AgNPs, leading to decrease of the protection against infection and consequent immune response. On the other hand, AgNPs may also interfere with lysosomal pH and cause excessive ROS production and impair the antimicrobial capabilities of zebrafish embryos.

Several studies have suggested that particle size strongly contributes to internalization and cell responses [3,54]. In the present study, it is possible that the AgNPs of ~50 nm in diameter exhibit both higher Ag deposition (Figure 2) and toxic effects than the AgNPs ~10 nm in diameter (Figure 4) because the specific size of the nanoparticles may incite particular cellular responses. For example, various studies have suggested that the highest cellular uptake occurs for nanoparticles that are approximately 30-50 nm in size due to the membrane-wrapping process [1,55]. Thermodynamically, particle sizes ranging between 30-50 nm are suitable for recruiting and binding to receptors, resulting in the successful activation of membrane wrapping. In contrast, the small-sized nanoparticles exhibit fewer interactions between the ligand and receptor. However, once the nanoparticle size is above 50 nm, it will possess a high affinity for a specific receptor and limit the binding of the receptor to other nanoparticles [1,55]. Therefore, the optimal particle size for internalization and the driving of dramatically adverse effects of nanoparticles could be 30–50 nm in diameter.

Previous studies have shown that AgNPs cause developmental delays and increase the occurrence of malformation [56,57]. In our results, we found that AgNPs increased developmental toxicity (Figure 5) and malformation phenotypes (Figure 6). We observed that the AgNP-induced malformation phenotypes were mainly yolk sac edema and heart failure. Interestingly, the larger size of AgNPs led to a higher percentage of yolk sac edema than the smaller size. Some studies have shown that AgNPs promote the occurrence of malformation via the influences of oxidative stress, cathepsin L, metallothionein, and endoplasmic reticulum calcium ATPase 1, which are malformation-related factors [57]. Moreover, other studies have found that AgNPs disrupt embryogenesis in medaka via the dysregulation of teratogenicity-related genes, including ctsL, tpm1, rbp, mt, and atp2a1 [57]. In the Medaka study, AgNPs significantly downregulated ctsL gene expression levels, which are known as lysosomal cysteine proteinases and are responsible for tumor promotion, bone resorption and growth regulation. In lower vertebrates, ctsL is expressed in the embryonic stages and participates in yolk proteolysis [58]. Therefore, AgNP-decreased ctsL expression levels may lead to teratogenicity and cause zebrafish embryo developmental toxicity.

In our study, we showed that the zebrafish yolk sac was stained with a higher intensity of Lysosensor in both the SAS and LAS groups (Figure 8A). Nanoparticles typically accumulate in lysosomes, and their impact on lysosomal function and autophagy have recently been proposed as an emerging mechanism of nanotoxicity [43,44,59]. Autophagy plays a critical physiological role in mitigating stress-mediated protein aggregation and clearance of damaged organelles [37]. However, growing evidence suggests that various nanoparticles activate autophagy, which may serve as a cellular protective mechanism against toxicity. In our previous study, we proved that AgNPs can disrupt lysosomal activity and cause autophagy dysfunction and apoptosis in mouse fibroblast cells [37,38]. Recent studies have also shown that AgNPs block the process of autophagy due to impairment of autophagosome-lysosome fusion, which results in autophagic defects and consequently triggers the induction of multiple cytotoxic pathways [39].

AgNPs are widely used in many industrial and daily sections and thus facilitate their release into the environment via wastewater. It has been estimated by simulative modeling studies that the concentrations of AgNPs in U.S. surface waters are between 0.09 and 0.43 ng/L, those in European surface waters are between 0.59 and 2.16 ng/L, and those in the river Rhine are between 40 and 320 ng/L [9]. Moreover, the behaviors of nanoparticles can be changed by environmental conditions. It has also been reported that organic matrixes such as humic acid in natural water could attenuate AgNPs induced toxicity [60]. We found both SAS and LAS suspended in E3 embryo medium presented visible precipitation when it lasted for more than 12 h (data not shown).

An interesting finding on the developmental toxicity in embryos showed that AgNPs suspended in E3 medium elicited mild or none toxic effects on body length and head trunk angle when compared with deionized water. One possible explanation is that nanoparticles in the E3 medium could form aggregates. In a previous study, the authors indicated that the dispersion of zinc oxide nanoparticle in E3 medium was significantly reduced and formed aggregates, leading to a size of particles that exceeded the defined nanoscale range (1–100 nm), which may have affected the penetration through the chorion [61]. E3 medium, a multivalent inorganic salt solution, has been reported to trigger the instability of nanoparticles [62]; thus, the high ionic strength solution could contribute to the aggregation of AgNPs as demonstrated previously [63] and in our current study.

Our experimental exposure concentration of AgNPs in this study may be much higher than actual environmental AgNP concentrations. Nonetheless, from the toxicology/ecotoxicology point of view, determination of the dose response relationship is a central dogma for the risk assessment and hazard classification. Our experimental designs are based on “OECD Test No. 236: Fish Embryo Acute Toxicity”. Determination of LC_50_ and the toxic mechanisms of AgNPs can provide essential information further for both environmental and human health risk assessment.

In summary, our results suggest that zebrafish embryos take up a certain amount of AgNPs. Interestingly, AgNPs ~50 nm in diameter showed a different potency from that of AgNPs ~10 nm in diameter. The LAS increased the pore size and penetrated the chorion, leading to an abundance of AgNP deposition. The accumulated AgNPs induced adverse toxic effects and developmental defects and increased the potential of specific malformations. AgNPs activate multiple cellular toxic pathways, including excessive ROS, activated lysosomal activity and even increased cell death. Taken together, we determined that AgNPs triggered size- and dose-dependent adverse effects in a zebrafish embryo model (Figure 9).

## 4. Materials and Methods

### 4.1. SAS/LAS Synthesis and Characterization

SAS and LAS were prepared by surface functional substitution from 20 nm and 50 nm citrate-coated AgNPs, respectively. The AgNPs were prepared by the NaBH_4_ reduction of AgNO_3_. Briefly, aqueous silver nitrate solution (1.5 mL, 20 mM) and sodium citrate solution (1.2 mL, 80 mM) were mixed with H_2_O (26.7 mL). Ice-cold aqueous NaBH_4_ solution (0.6 mL, 100 mM) was added dropwise under vigorous stirring. Following this process, the resultant colloidal AgNP suspension was stored at 4 °C and kept from light. The required size of the NPs was separated by centrifugation. Then, 40 mL 40 ppm prepared large (83 nm) and small (26 nm) silver nanoparticles were mixed with 100 mL of cysteamine, and added with 1 mL of sodium hydroxide (NaOH) to neutralize the ions released by cysteamine (the exact volume required depending on the concentration of AgNPs); if the added of cysteamine causes AgNPs to be reduced, the volume needs to be reduced. Then, the mixture was vigorously stirred at room temperature. After reacting for 2 h, the silver nanoparticles were purified by centrifugation and resuspend with deionized water. Characterization of the SAS and LAS was performed using TEM (JEOL Co., Akishima, Tokyo, Japan). SAS and LAS were examined after suspension in M.Q. water and deposition onto copper-coated carbon grids. TEM software was calibrated to measure the sizes of the SAS and LAS. The compositions of SAS and LAS were determined by EDX analysis (JEOL Co., Akishima, Tokyo, Japan). The hydrodynamic sizes, zeta potential, and polydispersity index (PDI) of SAS and LAS examined by DLS (Delsa™ Nano C, Beckman Coulter, Inc., Brea, CA, USA). The zeta potential of the SAS and LAS was analyzed in aqueous dispersion using a phase analysis light scattering (PALS) (Delsa™ Nano C, Beckman Coulter, Inc., Brea, CA, USA).

### 4.2. Preparation of Rhodamine 6G Conjugated SAS/LAS

The final product cysteamine AgNPs were further synthesized and mixed with Rhodamine B isothiocyanate (R6G) (Sigma-Aldrich, St. Louis, MO, USA). To 20 mL, 800 ppm AgNPs was added 2 mL of 1 mM R6G. The solution was stirred vigorously at room temperature for 2 h. The AgNPs were purified by centrifugation (depending on the size of the desired nanoparticle). The supernatant was removed and deionized water was used to redissolve the precipitate. The centrifugation step was repeated until the supernatant was clarified. Finally, the precipitate was dissolved back into deionized water to obtain R6G-conjugated SAS/LAS. R6G-conjugated SAS/LAS exhibited red fluorescence and we used them to observe the locations of the SAS and LAS in zebrafish embryo.

### 4.3. Atomic Absorption Spectrometry

To determine the concentration of the synthesized silver nanoparticles, atomic absorption spectrometry (AAS) was used to examine the amount of silver. A standard solution of AgNO_3_ was prepared and diluted to a proper concentration to make a standard curve. The sample was diluted into the range of the standard curve and then digested at 95 °C for 10 min. Then, the concentration was analyzed using a graphite furnace atomic absorption spectrometer (Perkin Elmer AAnalystTM 600, Waltham, MA, USA).

To quantify the distribution concentration of SAS and LAS retention in the embryos, after zebrafish embryo exposure to AgNPs (1 and 10 µg/mL), embryos were collected at 48, 72, and 96 hpf, and AAS was used to examine the cumulative amount of silver in the embryos. The collected embryos of the experimental and untreated groups were treated with 1.5 mL of nitric acid (69%) for 4 h at 50 °C, followed by another 4 h at 100 °C. Once fully digested, the samples were diluted with 2% nitric acid to an acceptable concentration (approximately 200- to 300-fold dilutions depending on the exposure dose) and then subjected to analysis using a graphite furnace atomic absorption spectrometer. Triplicate measurements of the absorption for the sample were taken for each exposure dose.

### 4.4. Scanning Electron Microscopy (SEM)

To analyze the chorion, we used SEM with the principle of BSE. The sample chorions were rinsed with PBS and then 20–25 µL of sample was siphoned to load onto the center of the glasses. Fifty microliters of 2.5% glutaraldehyde was added to every sample and followed by sealing. The glasses were swayed for 20 min at 4 °C. The glutaraldehyde was removed and the sample was rinsed with PBS for 3–5 min three times. To each plate was added 1.0% OsO_4_ (osmium tetroxide) (Sigma-Aldrich, St. Louis, MO, USA) to start the reaction for 40 min. Dehydration was operated by the following concentrations: 50%, 70%, 80%, 90%, 95% and 100% three times (using 0.5 mL of acetone and 3 min for every glass). Every glass was packed with filtrate paper and the samples were placed on the upper side. The samples were dried at the temperature critical point and coated with gold. Finally, the sample as well as the glass were loaded on the field-emission scanning electron microscope instrument (JSM-7001F, JEOL Co., Tokyo, Japan) at 28–30 kV to scope.

### 4.5. Zebrafish Maintenance and Egg Spawning

Zebrafish (Danio rerio) (Gendanio, AB line wild-type, New Taipei, R.O.C) in adulthood were maintained at 28.0 °C with a light cycle of 14 h light/10 h dark and fed brine shrimp twice per day and feed (containing 60% rough protein) once per day. All experiments on zebrafish were performed according to the guidelines of our institute (Guide for Care and Use of Laboratory Animals, National Cheng Kung University Medical College) and were approved by the Institutional Animal Care and Use Committee of National Cheng Kung University, Taiwan (Approval No.: 107133; Date: February 27, 2018). To obtain zebrafish embryos, male and female zebrafish were set up as pairs the night before mating in breeding tanks with a divider. The next morning, the zebrafish were stimulated with light, and the dropped eggs were collected, pooled, and rinsed with tap water and then deionized water several times. For the experiments, fertilized eggs were chosen and collected under a microscope, and the dead/unfertilized embryos were removed to avoid test solution pollution. All zebrafish experiments were performed in compliance with guidelines from the OECD Test Guidelines (TG203 and TG236) [64].

### 4.6. Zebrafish Embryo Acute Toxicity Test

The fish embryo acute toxicity (FET) test using zebrafish is used to examine toxicity in 96 h [65]. In our research, we modified some testing conditions to meet our endpoint. Fish embryos were collected and exposed to the test solution at 4 hpf, 28.0 °C (*n* = 30). For the embryo acute toxicity test, all the development takes place in milli Q water or AgNPs solution. The embryos were maintained in milli-Q water with AgNPs (LAS/SAS groups) or milli-Q water alone (Control group) during 4 hpf to the specific endpoint. The exposure solutions were renewed every 24 h to avoid aggregation of AgNPs. Then, the numbers of alive and dead embryos at 24, 48, 72, 96, and 120 hpf were counted, and photos were taken using a microscope with a mounted CCD camera (SMZ800, Nikon Instruments Inc., Melville, NY, USA).

### 4.7. Developmental Toxicity

We used healthy zebrafish to spawn embryos. We then washed the embryos with E3-media and egg water after collection. The used criteria came from prior study [2]. For developmental toxicity assay, the zebrafish embryos were exposed to AgNPs suspended both in deionized water and E3 medium, a culture medium usually applied to developmental study. The development of zebrafish embryos could be measured by their body length and head-trunk angle. The head-trunk angle is a mathematical function of zebrafish embryo development at 28.5 °C. The main purpose is to measure the angle between the head and trunk and then determine whether the development is normal, stimulated or inhibited. We then calculated the malformation rate and the kind of malformation, including some criteria indicated malformations, such as yolk sac edema, opaque yolk, pericardial edema and axial curvature. Additionally, we observed the malformations of bent axial, tail damage and jaw bulge.

### 4.8. ROS Measurement

ROS generation induced by different doses (0, 0.1, 1, 10 µg/mL) of LAS and SAS was measured. For sample preparation, 0.003% 1-phenyl-2-thiourea (PTU) (Sigma-Aldrich, St. Louis, MO, USA) was added to decrease the rate of pigment formation. Then, 48, 72, or 120 hpf, the reactive oxygen species (ROS) in zebrafish embryos was measured. Before measuring, 1 μg/mL DCFH-DA dye (Sigma-Aldrich, St. Louis, MO, USA) was added to each group (0, 0.1, 1, 10 µg/mL). After cultivating and avoiding light for 30 min under an ambient environment, the embryos were rinsed with deionized water several times and fixed on glass with 3% methylcellulose. Then, we observed the ROS generation with a confocal microscope (excitation/scattering = 488 nm/543 nm) and captured photos. An apical point was to analyze AgNPs-induced ROS in zebrafish embryos. Finally, the quantitative fluorescence was measured using the panoramic tissue somatic cell quantitative software TissueQuest (TissueGnostics, Vienna, Austria).

### 4.9. Lysosomal Activity Assay

LysoSensor^TM^ (Thermo Fisher Scientific Inc., Waltham, MA, USA) induces green florescence in a pH < 5 environment and distinguishes the activity of the cellular lysosome. Embryos at 4 hpf were exposed to AgNPs and 0.03% PTU was added to avoid fish pigment precipitation. At 48 hpf, the AgNP solution was removed and the embryos were rinsed with Milli-Q water 3 times. Finally, 10 µM Lysosensor kit was added, and the samples were incubated for 45 min at 28 °C. After cultivation, the solution was removed and the samples were rinsed with 1 mL of Milli-Q water 3 times. The fish were fixed with 3% methylcellulose and observed under a vertical fluorescence microscope (BX51, Olympus Co., Tokyo, Japan). The phenotypes of the lysosomes are red dots.

### 4.10. TUNEL Assay

To detect apoptotic cells in live embryos, we conducted a whole-mount TUNEL assay (Sigma-Aldrich, St. Louis, MO, USA). Embryos were exposed to different doses (0, 0.1, 1, 10 µg/mL) of LAS and SAS at 4 hpf, and the exposure method was the same as the zebrafish embryo acute toxicity test. The endpoint of the experiment was at 72 hpf. The exposure solution containing AgNPs was placed in a 12-well plate and then washed three times with deionized water, and the embryos were collected in an Eppendorf microcentrifuge tube in the test concentration group. Embryos were fixed with 4% paraformaldehyde at room temperature for 1 h. After fixation for 1 h, we washed the embryos with PBS on a shaker (40 rpm) for 5 min, which was repeated twice. Then, we incubated the embryos with blocking buffer on a shaker for 30 min. After 30 min, the embryos were washed with PBS on a shaker (40 rpm) for 5 min, which was repeated twice. Then, the embryos were incubated with permeabilization solution (0.1% Triton X-100 in 0.1%) on ice for 30 min. After that, we washed the embryos with PBS on a shaker (40 rpm) for 5 min twice. Finally, the TUNEL reaction mixture (labeling solution:enzyme solution = 9:1) was incubated at 37 °C for 1 h in a water bath and finally observed with a fluorescence microscope.

### 4.11. Statistics

All experiments were conducted in triplicate. We used statistical software to analyze the data, such as IBM SPSS statistics 22 and Excel. Statistical analyses were carried out using the two-tailed Student’s *t*-test for comparisons between the means or using one-way ANOVA. Differences were considered to be significant when *p* < 0.05. All data are presented as the means± SD.

## Figures and Tables

**Figure 1 ijms-21-02864-f001:**
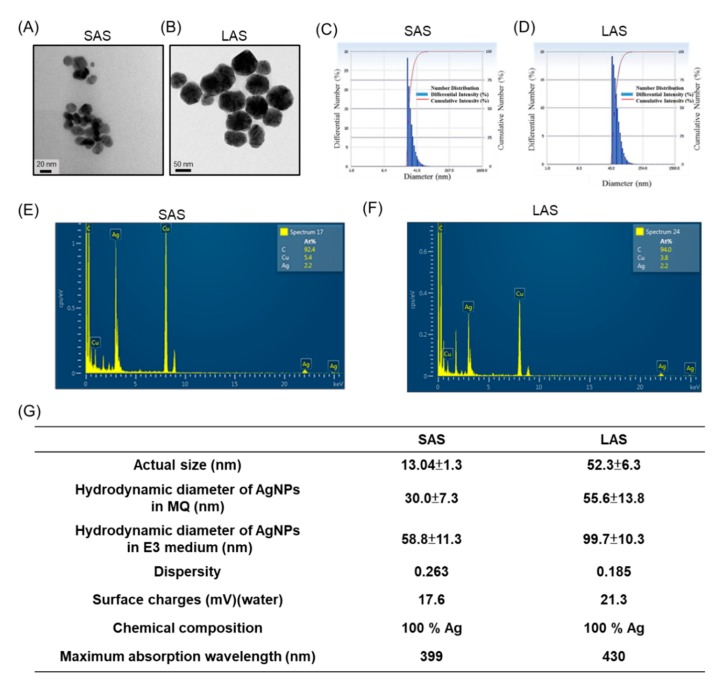
The physical-chemical properties of the SAS and LAS. (**A**,**B**) Transmission electron microscopy (TEM) images of SAS and LAS. (**C**,**D**) The diameter and number distribution of SAS/LAS were measured by dynamic light scattering. (**E**,**F**) The composition of SAS and LAS were analyzed via energy-dispersive X-ray spectroscopy. (**G**) Measurement of physical-chemical properties, including primary size, hydrodynamic diameter, dispersity, surface charges, chemical composition and maximum absorption wavelength.

**Figure 2 ijms-21-02864-f002:**
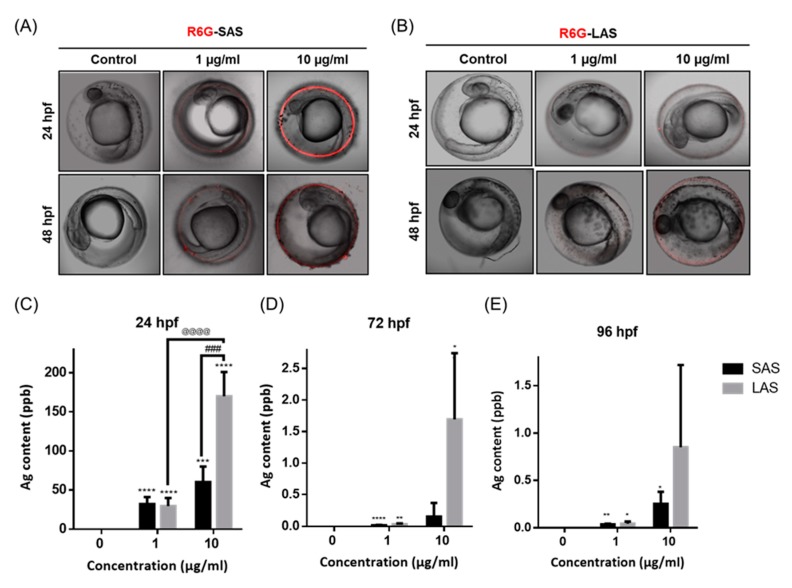
The metal accumulation of SAS and LAS. Zebrafish embryo were exposed to 1, 10 µg/mL R6G conjugated (**A**) SAS (**B**) LAS at 24–48 hpf. Red fluorescence signals represent deposition of AgNPs. The retention of silver in embryos at (**C**) 24, (**D**) 72, (**E**) 96 hpf. The level of silver ion was detected by atomic absorption spectrophotometer analyses. The experiment was performed thrice using 30 embryos each. The values are presented as the mean ± SEM. Values significantly different from the control are indicated by asterisks (one-way ANOVA, followed by a *t*-test: * *p* < 0.05, ** *p* < 0.01, *** *p* < 0.001 and **** *p* < 0.0001). The significantly difference between SAS and LAS are indicated by pound sign (one-way ANOVA, followed by a *t*-test ### *p* < 0.001). The significantly difference between LAS 1 µg/mL and LAS 10 µg/mL groups are indicated by the at sign (one-way ANOVA, followed by a *t*-test ^@@@@^
*p* < 0.0001).

**Figure 3 ijms-21-02864-f003:**
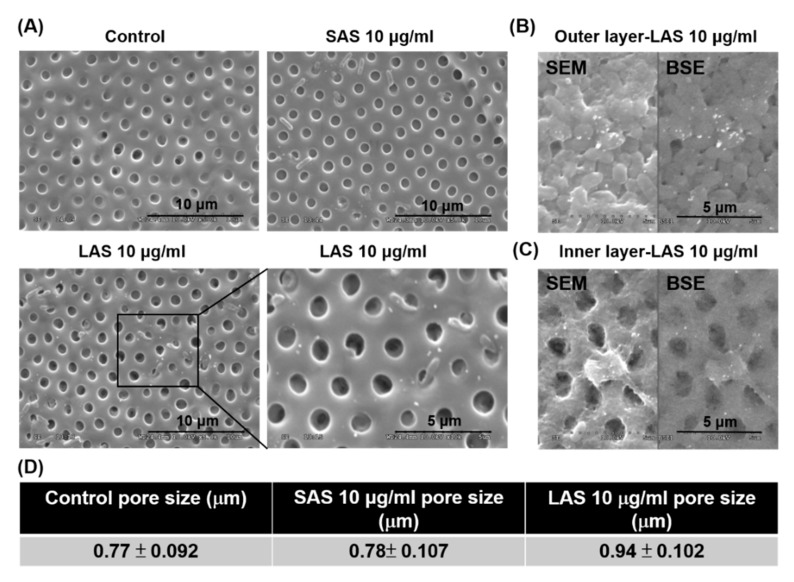
The chorion inner/outer membrane of SAS/LAS-treated zebrafish embryos. Zebrafish embryos were observed via SEM after exposure to (**A**) 10 µg/mL SAS and LAS at 48 hpf. The backscattered electrons of the (**B**) outer layer and (**C**) inner layer membrane. The white dots indicate the accumulation of LAS. (**D**) The diameter of the chorion inner membrane pores was approximately 0.77 μm for the control, 0.78 μm for SAS and 0.94 μm for LAS.

**Figure 4 ijms-21-02864-f004:**
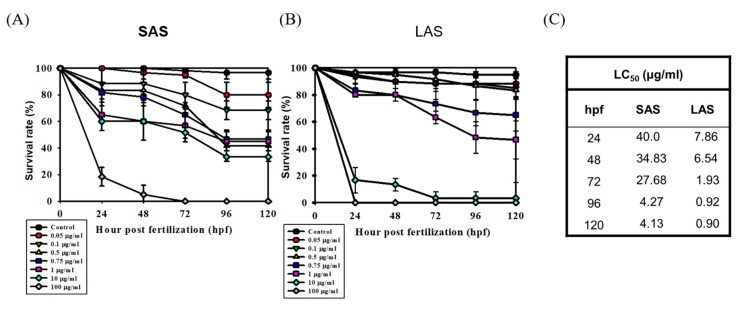
The toxic effects of SAS/LAS. The mortality of embryos treated with 0, 0.05, 0.1, 0.5, 0.75, 1, 10, or 100 µg/mL (**A**) SAS or (**B**) LAS at 24, 48, 72, 96, 120 hpf. (**C**) LC_50_ of SAS/LAS-treated embryos. The lower LC_50_ of LAS indicates that the toxicity of LAS is higher than that of SAS at 24, 48, 72, 96, and 120 hpf. The LC_50_ of SAS/LAS was calculated by IBM SPSS 22 statistics. The mortality was performed from 3 replicate trials using 30 embryos at each dose. Error bars present the SEM of the mean.

**Figure 5 ijms-21-02864-f005:**
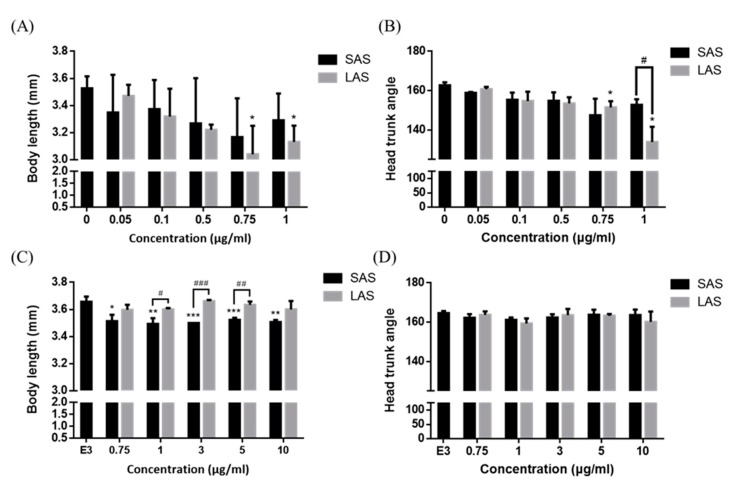
The developmental toxicity of zebrafish embryos. The developmental toxicity of SAS/LAS suspension in deionized water (**A**,**B**) and suspension in E3 embryo medium (**C**,**D**) were evaluated via body length and head-trunk angle. Zebrafish embryos were treated with SAS and LAS suspension in deionized water (0, 0.05, 0.1, 0.5, 0.75, 1 μg/mL) and E3 medium (0, 0.75, 1, 3, 5, 10 μg/mL) for 72 hpf to measure the body length and head-trunk angle. The values are presented as the mean ± SEM. Values that are significantly different from the control are indicated by asterisks. (one-way ANOVA, followed by a *t*-test: * *p* < 0.05, ** *p* < 0.01 and *** *p* < 0.001). LAS control versus exposed LAS groups). Values that are significantly different between SAS and LAS is indicated by the pound sign (one-way ANOVA, followed by a *t*-test: # *p* < 0.05, ## *p* < 0.01 and ### *p* < 0.001).

**Figure 6 ijms-21-02864-f006:**
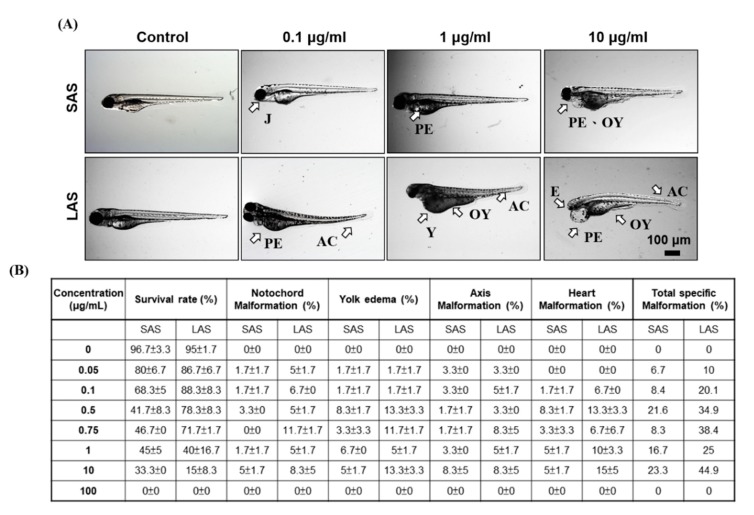
Malformation of zebrafish embryos exposed to LAS and SAS. (**A**) The SAS/LAS-treated embryos (0, 0.1, 1, 10 μg/mL) revealed different malformed phenotypes at 96 hpf. J, jaw malformation; PE, pericardial edema; OY, opaque yolk; AC, axial curvature; Y, yolk sac edema; E, eye malformation. (**B**) Quantification of the survival rate and specific malformation rate, including notochord malformation, yolk edema, axis malformation, and heart malformation. The total malformation indicates the sum of the specific malformation rate. The values are presented as the mean ± SEM.

**Figure 7 ijms-21-02864-f007:**
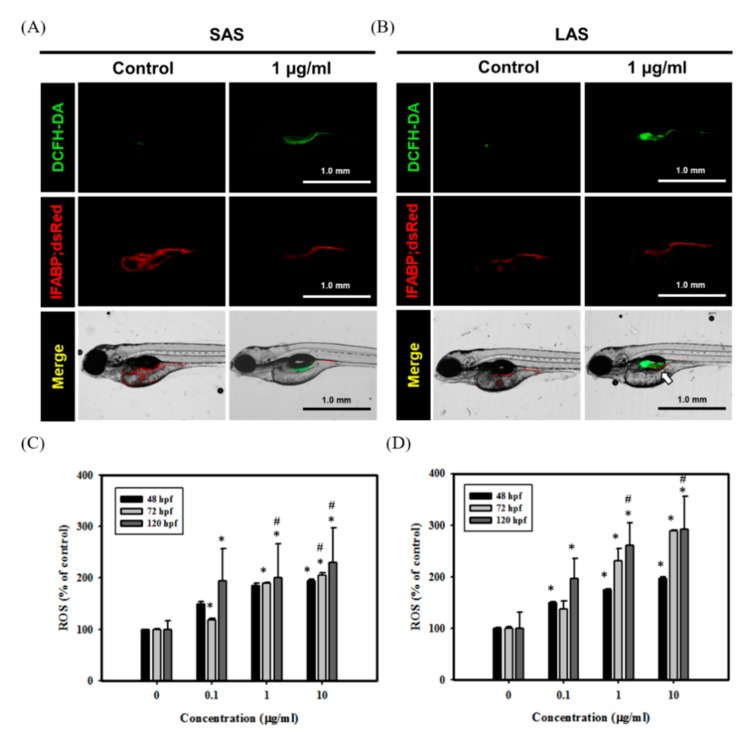
SAS/LAS induced excessive ROS in zebrafish embryos and damaged the intestinal region. Transgenic zebrafish (Tg(IFABP;dsRed)) embryos were treated with 1 µg/mL of (**A**) SAS and (**B**) LAS for observing the intestinal region and ROS production at 120 hpf. The ROS levels were analyzed via DCFH-DA assay. GFP fluorescence signals indicate ROS. RFP fluorescence signals represent the intestinal region. The white arrow indicates the excessive ROS in LAS group. ROS production of (**C**) SAS- (**D**) LAS-treated groups (0, 0.1, 1, and 10 µg/mL) were quantified at 48, 72 and 120 hpf. Quantification of ROS levels was performed with TissueQuest software. The values are presented as the means ± SEM. Values that are significantly different from the control are indicated by asterisks (one-way ANOVA, followed by a *t*-test: *, *p*< 0.05, control versus exposed groups. #, *p*< 0.05).

**Figure 8 ijms-21-02864-f008:**
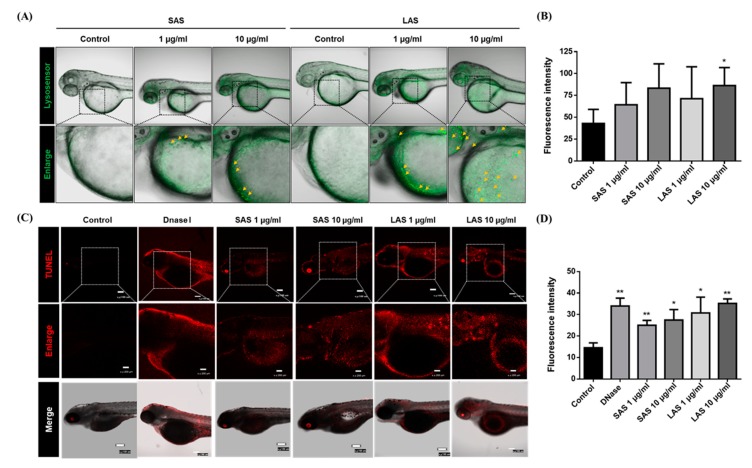
Effect of SAS/LAS on lysosomal activity and apoptosis. (**A**) 4 hpf zebrafish embryos were treated with SAS, LAS (0, 1, 10 μg/mL) and stained by Lysosensor, at 48 hpf. The green color and arrows represent upregulation of lysosomal activity. (**B**) The quantification of lysosensor fluorescence intensity (**C**) The zebrafish embryos treated SAS/LAS (0, 1, 10 μg/mL) were conducted at 72 hpf. Dnase I is the positive control of TUNEL assay. Red color represents apoptotic cell. Apoptotic cell mainly distributed to axis, yolk sac and head region. The scale bars in top and bottom panel (TUNEL and Merge) and middle panel (Enlarge) represents 100 μm and 200 μm respectively. (**D**) The quantification of TUNEL fluorescence intensity. The values are presented as the mean ± SEM. Values that are significantly different from the control are indicated by asterisks (one-way ANOVA, followed by a *t*-test: * *p* < 0.05 and ** *p* < 0.01).

**Figure 9 ijms-21-02864-f009:**
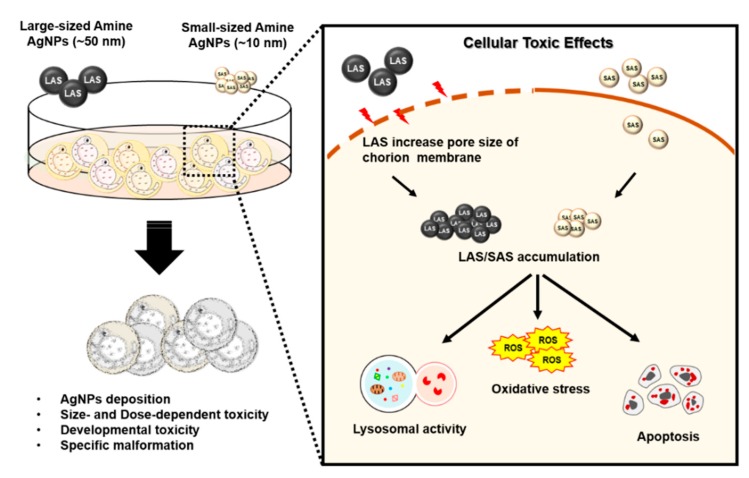
Illustration of SAS-/LAS-induced toxic effects in zebrafish embryos. AgNPs triggered size- as well as dose-dependent toxicity and accumulated in the embryos. The deposition of AgNPs increased the number of developmental defects and the induction of specific malformations. Mechanically, AgNPs activate various cellular toxic signaling pathways, including increased ROS, lysosomal activity and apoptosis. Interestingly, LAS showed different behavior from SAS. LAS increased the pore size and penetrated the chorion, resulting in an abundance of AgNP deposition.

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
