# Peer review of "The Effect of the Chorion on Size-Dependent Acute Toxicity and Underlying Mechanisms of Amine-Modified Silver Nanoparticles in Zebrafish Embryos"

_ijms, 2020, doi:10.3390/ijms21082864_

Round 1

Reviewer 1 Report

The study provides the toxicity information of AgNPs in zebrafish embryo. The presented data is interesting, however the experimental design cannot reflect the real situation in the environment/ or mammalian cases. The major questionable issue is the experimental exposure concentration of AgNP. The dosage authors using are far too high (ug/ml), authors should clarify that concentration or should mention such limitation in the MS.

What is the efficiency of R6G conjugated with SAS or LAS, are they the same? Any data to support that? If it is different, there is no point for further comparison. material 4.2 mentioned that the dye amount is depends on AgNP concentration and saturation signals. So, it seems to me that the added dye is different between SAS and LAS. ..then the signal brightness mentioned in the result is not appropriated.

It seems that the error bar of 10ug/ml was big. I would like to know if there are sig. different between 1 and 10ug/ml in each cases (2C-2H). Furthermore, Figure 2A/B, 1ug/ml exposure SAS seems to have higher signal than LAS. Please provide quantified data.

Figure 4. Mortality of embryos: SAS. survival rate dropped to 0 at 72 hpf onwards, then why the authors have the data in figure  2D and 2E. those result were from the dead fish bodies? same as 6A, where comes the 10ug/ml 96hpf? However the table in Fig.6 showed 33% survival rate, which is totally diff from 4A. Please check.

To summarize, based on the table in figure 6, the survival rate of 1ug/ml exposure of either SAS or LAS is ~40%. Thus, the imaging, or phenotypic data are selected from these 40% embryos, which may not be representative (besides it is not environmental relevant). Please clarify.

Figure 8 is not clear enough. Please select better images. Furthermore, quantified data is necessary. Since the cell death data is solely based on the dye imaging. It is better to have one simple p53 PCR data to increase the merit.

Regarding figure 9 and discussion. It is interesting the LAS increase the pore size, and i think authors should discuss the possibility of increasing pore size will reduce the protection, and such ROS, lysosomal activity upregulation may due to infection or induced immune response.

Minor:
Figure 2. It is better to put SAS and LAS data in one bar chat for comparison (e.g. 2 C and 2F together etc..) it is the same case in figure 5. putting them in one graph will be better for comparison.

Author Response

Our point-by-point response is as follows:

Point 1: The study provides the toxicity information of AgNPs in zebrafish embryo. The presented data is interesting, however the experimental design cannot reflect the real situation in the environment/ or mammalian cases. The major questionable issue is the experimental exposure concentration of AgNP. The dosage authors using are far too high (ug/ml), authors should clarify that concentration or should mention such limitation in the MS.

Response 1: Thank you very much for your valuable comments. Our experimental designs are based on “OECD Test No. 236: Fish Embryo Acute Toxicity”. The following description is an excerpt from TG236. “Five concentrations of the test chemical spaced by a constant factor not exceeding 2.2 are required to meet statistical requirements”. The highest concentration test should preferably result in 100% lethality. Moreover, in toxicology research, it applies the concept of dose descriptor, such as median lethal concentration (LC50) to estimate the risk for environment and human health. Indeed, the experimental exposure concentration of AgNPs cannot reflect the real situation in the environment. Thus, we discuss the limitation in the discussion section. The following statement has been given in the “Discussion” section. “AgNPs are used in many industrial and daily necessities thus increase the possibility of AgNPs entering the environment via wastewater. It has been estimated by the simulative modeling studies that the concentration of AgNPs in U.S. surface waters are between 0.09 and 0.43 ng/L, in European surface waters are between 0.59 and 2.16 ng/L and in river Rhine are between 40 to 320 ng/L [1]. It has also been reported that the organic matrix such as humic acid in natural water could attenuate AgNPs induced toxicity [2]. Thus, our current experimental exposure concentration of AgNP is much higher than the real situation in the environment. Nonetheless, from the toxicology/ecotoxicology point of view, determination of the dose response relationship is a central dogma for the risk assessment and hazard classification. Our experimental designs are based on “OECD Test No. 236: Fish Embryo Acute Toxicity”. Determination of LC50 and the toxic mechanism of AgNPs can provide essential information for further risk estimation both in environment and human being.”

Point 2: What is the efficiency of R6G conjugated with SAS or LAS, are they the same? Any data to support that? If it is different, there is no point for further comparison. material 4.2 mentioned that the dye amount is depends on AgNP concentration and saturation signals. So, it seems to me that the added dye is different between SAS and LAS. ..then the signal brightness mentioned in the result is not appropriated.

Response 2: Thank you for pointing out the important concern. Actually, the efficiency of R6G conjugated with SAS or LAS is different due to their different surface area under the same weight unit. Thus, it is inappropriate to quantify and compare the intensity of R6G-SAS and R6G-LAS. Therefore, we have modified the description in 2.2 section and 4.2 section as following: 2.2. Distribution of the AgNPs in developing zebrafish embryos To investigate the distribution of AgNPs in zebrafish embryos, we used rhodamine B isothiocyanate (R6G)-conjugated SAS and LAS to provide dynamic imaging of zebrafish embryos. Figure 2A indicates that R6G-SAS accumulated in the outer layer of the chorion of the zebrafish embryos at 24 hpf and 48 hpf. Conversely, the LASexposed group tends to penetrate into the zebrafish embryo at 48 hpf (Figure 2B). 4.2. Preparation of rhodamine 6G conjugated SAS/LAS The final product cysteamine AgNPs were further synthesized and mixed with rhodamine B. To 20 ml, 800 ppm AgNPs was added 2 ml of 1 mM rhodamine B. The solution was stirred vigorously at room temperature for 2 hours. The AgNPs were purified by…….to obtain rhodamine 6G-conjugated SAS/LAS. Rhodamine 6Gconjugated SAS/LAS exhibited red fluorescence and we used them to observe the locations of the SAS and LAS in zebrafish embryo.

Point 3: It seems that the error bar of 10 ug/ml was big. I would like to know if there are sig. different between 1 and 10 ug/ml in each cases (2C-2H). Furthermore, Figure 2A/B, 1ug/ml exposure SAS seems to have higher signal than LAS. Please provide quantified data.

Response 3: Thank you for raising this concern. Yes, the error bar of 10 ug/ml was big in 72 and 96 hpf groups, leading to a non-significant results between the 1 and 10 µg/ml groups. Whereas, the error bar of 10 ug/ml in 24 hpf group was reasonable and we still can find the significance between the 1 and 10 µg/ml groups. One of the possible reasons is because the dechorionation of the embryos at 72 and 96 hpf. Thus, the silver content decreased dramatically. In the figure 2A/B, we examined the distribution of AgNPs via R6G-AgNPs. The aim of the R6G conjugated SAS and LAS are mainly to locate the region of SAS/LAS deposition. The results show the R6G specific pattern around the chorion, particular at SAS data. Because the efficiency of R6G conjugated with SAS or LAS is different, we could not quantify the silver content by images. Instead, we performed the atomic absorption spectroscopy assay (AAS) to compare the accumulative level of SAS/LAS in zebrafish (2C-E). According to our data, the deposition of LAS is higher than SAS.

Point 4: Figure 4. Mortality of embryos: SAS. survival rate dropped to 0 at 72 hpf onwards, then why the authors have the data in figure 2D and 2E. those result were from the dead fish bodies? same as 6A, where comes the 10ug/ml 96hpf? However, the table in Fig.6 showed 33% survival rate, which is totally diff from 4A. Please check.

Response 4: Regarding the concern about survival rate dropped to 0 at 72 hpf onwards in Fig. 4, and the data in figure 2D and 2E. I think it’s a misunderstanding from the reviewer. Actually the survival rate dropped to 0 at 72 hpf is under the concentration of 100 µg/ml, but not 1 and 10 µg/ml. Survival rate of SAS 1, 10 µg/ml at 72 hpf are about 50-60%, and 40-50 % respectively. We collected the alive fishes to conduct the atomic absorption spectroscopy assay (Figure 2C-E) and malformation assay (Figure 6A). Although LAS (10 µg/ml) cause extreme toxicity of zebrafish embryo at 72 hpf, we still can collect about 10% of alive fishes to evaluate the accumulative level of LAS. In figure 2A, the survival rate of 10 µg/ml SAS groups at 96 hpf are 46% and this result is similar to Fig.6B.

Point 5: To summarize, based on the table in figure 6, the survival rate of 1 ug/ml exposure of either SAS or LAS is ~40%. Thus, the imaging, or phenotypic data are selected from these 40% embryos, which may not be representative (besides it is not environmental relevant). Please clarify.

Response 5: Yes, the imaging or phenotypic data are selected from these 40% survival embryos treated with 1 ug/ml SAS or LAS. However, we thought it is still representative owing to the reason that dead embryos were not included in the calculation of malformation rate. In general, the incidence of lethality and malformations were applied to represent zebrafish embryo acute and developmental toxicity and the deformity rates are calculated from the survival zebrafish embryo/larvae. Regarding the concern of environmental relevant, we have addressed it in the above mentioned “Response to reviewer’s No. 1 question” and also in the “Discussion” section.

Point 6: Figure 8 is not clear enough. Please select better images. Furthermore, quantified data is necessary. Since the cell death data is solely based on the dye imaging. It is better to have one simple p53 PCR data to increase the merit.

Response 6: Thank you very much for your valuable advices. We have improved the images (shown in figure 8A, 8C) and also quantified the images with fluorescence intensity as indicated in figure 8B and 8D. Regarding the cell death data, TUNEL assay is generally accepted as a useful marker of apoptosis. Apoptotic cell death can be triggered by both p53-dependent and p53-independent pathways. Detail mechanism and regulation of cell death pathways under environmental stress is a complicate issue. We appreciate the reviewer’s concern, and will take this issue into consideration in our future study.

Point 7: Regarding figure 9 and discussion. It is interesting the LAS increase the pore size, and i think authors should discuss the possibility of increasing pore size will reduce the protection, and such ROS, lysosomal activity upregulation may due to infection or induced immune response.

Response 7: Thank you very much for the constructive suggestions. We totally agree with the reviewer’s opinion, and have provided information regarding the possibility of increasing pore size will reduce the protection against infection, leading to immune response and consequent ROS generation and increased lysosomal activity. The following statement has been given in the “Discussion” section. “More interestingly, we found the increasing ROS level and lysosomal activity in zebrafish embryo exposed to AgNPs (Figure 7 and 8A). Lysosome is the definitive antimicrobial organelle [61]. It stands a crucial position in host–pathogen interactions, by being both targeted by pathogens and serving as a major mechanism for killing intracellular invaders [62]. Previous study indicated that the change of lysosomal enzymes activity may control microbial invaders in infected prawns [63]. The induction of ROS and lysosomal activity in zebrafish embryo could be attributed, on one hand, by the increasing pore size caused by AgNPs, leading to decrease of the protection against infection and consequent immune response. On the other hand, AgNPs may also interfere with lysosomal pH and cause excessive ROS production and impair the antimicrobial capabilities of zebrafish embryos.”

Point 8: Minor: Figure 2. It is better to put SAS and LAS data in one bar chat for comparison (e.g. 2 C and 2F together etc..) it is the same case in figure 5. putting them in one graph will be better for comparison.

Response 8: Thank you very much for the constructive suggestions. We have reconstructed Figure 2 and Figure 5 for better comparison of the SAS and LAS data.

Reference:

1. McGillicuddy, E., et al., Silver nanoparticles in the environment: Sources, detection and ecotoxicology. Science of The Total Environment, 2017. 575: p. 231-246.

2. Caceres-Velez, P.R., et al., Humic acid attenuation of silver nanoparticle toxicity by ion complexation and the formation of a Ag(3+) coating. J Hazard Mater, 2018. 353: p. 173-181.

Reviewer 2 Report

The TEM images of NPs must be modify because the magnification is different, and the measure is not correct. It's necessary to test the aggragation state of nanoparticles in the E3 media.

Author Response

Point 1: The TEM images of NPs must be modify because the magnification is different, and the measure is not correct. It's necessary to test the aggragation state of nanoparticles in the E3 media.

Response 1: Thank you for pointing out the mistakes. We have changed the TEM images of AgNPs in Figure 1A/B accordingly. Regarding the E3 media, we applied this media only for cleaning eggs before performing the developmental toxicity experiments as we described in the “Materials and methods” section 4.7. All the experiments of nanoparticles exposure were conducted in Milli-Q water, and that is the reason why we only provided the hydrodynamic diameter suspended in Milli-Q water in figure 1G.

Round 2

Reviewer 1 Report

The authors have answered my concerns. I do not have further questions on this MS.

Author Response

Response to Reviewer 1 Comments

Point 1:The authors have answered my concerns. I do not have further questions on this MS.

Response 1: Thank you very much for your valuable and helpful suggestion. 

Reviewer 2 Report

Thank you for tour revision, but I need a clarification on embryo toxicity tests; in your answer I read that the tests are conducted in milli Q water, I would like to understand if all the development takes place in milli Q water or the embryo develops in the absence of nanoparticles and when each single phase is reached, the nanoparticles are added.

Author Response

Response to Reviewer 2 Comments

Point 1: Thank you for tour revision, but I need a clarification on embryo toxicity tests; in your answer I read that the tests are conducted in milli Q water, I would like to understand if all the development takes place in milli Q water or the embryo develops in the absence of nanoparticles and when each single phase is reached, the nanoparticles are added.

Response 1: Thank you very much for your comments. In our experiments, all the development takes place in milli Q water or AgNPs solution. The embryos were maintain in milli-Q water with AgNPs (LAS/SAS groups) or milli-Q water alone (Control group) during 4 hpf to the specific endpoint. The exposure solutions were renewed every 24 hrs to avoid aggregation of AgNPs.

Round 3

Reviewer 2 Report

Dear authors
development experiments conducted in milliQ water are not adequate to test the possible effects of nanomaterials or nanoparticles on embryos. It's therefore necessary to carry out the experiments in E3 medium, it will also be necessary to evaluate the aggregation state of the nanoparticles in the culture medium.
